# Colorectal cancer trends in Chile: A Latin-American country with marked socioeconomic inequities

**Susana Mondschein**[1,2]*, **Felipe Subiabre**[1], **Natalia Yankovic**[3], **Camila Estay**[4], **Christian Von Mühlenbrock**[4,5], **Zoltan Berger**[4]

**1** Industrial Engineering Department, Universidad de Chile, Santiago, Chile, **2** Instituto Sistemas Complejos de Ingeniería, Santiago, Chile, **3** ESE Business School, Universidad de los Andes, Las Condes, Chile, **4** Department of Medicine, Gastroenterology Section, Hospital Clínico de la Universidad de Chile, Santiago, Chile, **5** Internal Medicine Department, Centro Enfermedades Digestivas, Clínica Universidad de los Andes, Las Condes, Chile

* susana.mondschein@uchile.cl

**Data Availability Statement:** The data underlying the results presented in the study is of public access at DEIS (https://deis.minsal.cl) This is third-

## Abstract

### Introduction

Colorectal cancer (CRC) is the third most frequent malignant disease in the world. In some countries with established screening programs, its incidence and mortality have decreased, and survival has improved.

### Aims

To obtain reliable data about the epidemiology of CRC in Chile, we analyzed the trends in the last ten years and the influence of observable factors on survival, including a nationwide health program for CRC treatment access (GES program).

### Methods

Publicly available data published by the Ministry of Health and National Institute of Statistics were used. Data were obtained from registries of mortality and hospital discharges, making follow-up of the individuals possible. Crude and age-standardized incidence and mortality rates were calculated, and individual survival was studied by constructing Kaplan–Meier curves. Finally, a Cox statistical model was established to estimate the impact of the observable factors.

### Results

We found 37,217 newly identified CRC patients between 2008 and 2019 in Chile, corresponding to 103,239 hospital discharges. In the same period, 24,217 people died of CRC. A nearly linear, steady increase in crude incidence, mortality and prevalence was observed. CRC incidence was the lowest in the North of the country, increasing toward the South and reaching a maximum value of 34.6/100,000 inhabitants/year in terms of crude incidence and 20.7/100,000 inhabitants/year in terms of crude mortality in the XII region in 2018. We found

party data and the authors report no special access privileges. We have also uploaded the database to a public repository to facilitate its access: (https://datos.uchile.cl/dataset.xhtml?persistentId=doi:10.34691/FK2/WUQXZP)".

**Funding:** This work was supported by the Complex Engineering Systems Institute [Centro Basal ANID / AFB180003]. Felipe Subiabre was financially funded by Agencia Nacional de Investigación y Desarrollo / Programa de Becas / Doctorado Nacional 21200869. The funders had no role in study design, data collection and analysis, decision to publish, or preparation of the manuscript.

**Competing interests:** The authors have declared that no competing interests exist.

that older patients had lower survival rates, as well as men compared to women. Survival was significantly better for patients with private insurance than those under the public insurance system, and the treating hospital also played a significant role in the survival of patients. Patients in the capital region survived longer than those in almost every other part of the country. We found no significant effect on survival associated with the GES program.

## Conclusions

We found important inequalities in the survival probabilities for CRC patients in Chile. Survival depends mainly on the type of insurance, access to more complex hospitals, and geographical location; all three factors correlated with socioeconomic status of the population. Our work emphasized the need to create specific programs addressing primary causes to decrease the differences in CRC survival.

## 1. Introduction

Colorectal cancer (CRC) represents up to 10% of cancers diagnosed worldwide each year and is the second and third most common in women and men, respectively [1]. CRC is related to lifestyle, and its incidence is increasing around the world [2]. Incidence rates vary according to geographic area, with the highest levels in developed countries and lower levels in developing countries [3]. It is estimated that by 2035, there will be 2.5 million new cases diagnosed, and the incidence of CRC in Latin America by 2030 will increase by 60%, with a total of 396,000 new cases per year [4]. The total number of deaths attributed to CRC is projected to increase in 60% and 71.5% in the colon and rectum, respectively, between 2013 and 2035, in part due to population growth and aging [5].

Reliable data on the statistics of CRC in Chile are relatively scarce, and most of the national investigations have focused on mortality rates, all of which have shown an upward trend over the years [6–8]. In fact, the latest publication reports that crude mortality for CRC in Chile for 2016 was 9.18 per 100,000 people, increasing more than 20% between 2000 and 2016 [8].

There are no publications regarding CRC incidence and survival rates. Official statistics (DEIS, MINSAL) accurately register the number of hospital discharges, but there is no individualized number of cases with first diagnosis of CRC. Newly diagnosed CRC cases and variations because of COVID 19 pandemic were analyzed using a simulation model in a recent work, demonstrating a projected backlog in disease diagnosis which results in a worse stage distribution leading to an increase in mortality [9].

Chile has a particular geography with nearly 5,000 km longitudinal extension with varying climatic conditions, nutritional habits, and ethnic composition. Epidemiological differences have been reported in gastric and gallbladder cancer, showing a higher frequency in *the Mapuche* population, concentrated in southern Chile [10–12]. The geography of Chile and its regions is depicted in the (Fig 1 in S4 Appendix, left map).

The most impressive statistic is that the mortality/incidence ratio is high in Latin America and the Caribbean, where six out of ten patients die, while in the USA, it is three out of ten [13]. Some plausible explanations refer to different health care systems and early screening strategies, which are advanced in the USA and practically absent in Latin America. Notably, extreme inequality in social factors such as education and income has caused poor outcomes in cancer survival. Moreover, the personal and familial economic burden of CRC diagnosis strongly depend on health insurance.

Several publications [14–16] are dedicated to analyze the survival of CRC in different specialized centers, depending on the phase of the disease. However, we have no information on the lethality of CRC at the national level. Survival depends on the diagnosis of the disease in the earliest stage possible and on rapid access to adequate treatment.

The Chilean health care system is a hybrid of public and private providers and insurances consisting of i) Fondo Nacional de Salud (FONASA–National Health Fund), which is public insurance for 78% of the Chilean population; ii) private health care insurers (ISAPREs) for 14% of the population; and iii) the Military and Police Forces' health system that represents 2.8% of the population. Privately insured patients can only access private providers (with a variety of coverages), while FONASA patients may access public and some private providers, depending on their income level, with different copayments (see in Table 1 in S1 Appendix, for the description of groups of FONASA patients).

In 2000, Chile introduced an extensive reform of the healthcare system, aimed at achieving a more equitable and fairer system for all citizens, with a law that took effect in 2004 [17]. This reform, initially known as AUGE (Acceso Universal de Garantías Explícitas) and later renamed the GES plan (Garantías Explícitas en Salud), consists of a set of guarantees regarding access, quality, opportunity, and financial protection for all Chilean citizens under FONASA or ISAPREs health insurances diagnosed with any of an established set of diseases (currently 84 conditions). Treatment for CRC was included in the GES program in 2014. All privately insured patients have a special coverage for high-cost diseases and treatments that is activated on top of the GES guarantees.

To determine a median-to-long term strategy at the national level, it is necessary to know the real situation of the disease. The primary aim of our present study was to obtain reliable information on annually diagnosed new CRC cases in Chile, i.e., the incidence of CRC, and to describe their epidemiological characteristics and geographical distribution. The secondary aim was to analyze the influence of several observable factors on patient survival, including some socioeconomic aspects, namely, differences in health insurance, the introduction of the GES program, and specific hospitals' characteristics.

## 2. Materials & methods

### 2.1 Ethics statement

This work used publicly available data at the Chilean Ministry of Health through the Departamento de Estadísticas e Información de Salud (DEIS). All data are protected, and personal information is anonymized. Therefore, no consent from participants was required.

### 2.2 Data description

We used the national registry of all inpatient discharges from hospitals in Chile, considering both the public and private sectors, for the period between 2001 and 2019. The database has 39 fields, including primary and secondary diagnosis, sex, age, ethnicity, health insurance, hospital, region of residency, length of stay, and condition at discharge.

We constructed a treatment database considering all patients for whom we had a first registry between 2009 and 2018 with a diagnosis code associated with CRC, resulting in 37,217 patients, corresponding to 103,239 different hospitals discharge episodes with a mean of 2.8 (std 4.0) hospitalizations per patient. We remark that we consider the date of the first treatment as the time of diagnosis. Thus, we construct the database with patients whose first registry was observed starting from 2009 to make sure they did not receive treatment for at least 8 years (from 2001 to 2008), and therefore can be confidently classified as newly diagnosed patients.

In a similar way, using the national death registry, we constructed a death database considering all deaths between 2009 and 2018 for patients with a diagnosis of CRC, either identified in the treatment database (15,635 patients) or without any CRC hospital discharge from 2001 onwards (6,099 patients), leading to a total of 21,734 deaths.

For both the treatment and death databases, the principal and related diagnoses were defined using the International Classification of Diseases, 10th Revision (ICD-10) codes. We therefore identified three relevant subsets of the ICD-10 codes (see S2 Appendix). The first one (Table 2 in S2 Appendix), called principal death and treatment codes, includes those directly identifiable as death or treatment due to CRC. The second group (Table 3 in S2 Appendix), called principal death codes and related treatment codes, is a subset of codes identified with deaths directly identifiable as caused by CRC but are related to CRC for the treatment database. Finally, the third group (Table 4 in S2 Appendix), called related codes, includes ICD-10 codes that do not by themselves indicate a CRC diagnosis but can be confidently linked to it when paired with a principal code. These are relevant to correctly distinguish the patients' causes of death (whether by CRC or an unrelated cause) and identify CRC survival times. Fig 1 summarizes the inclusion and exclusion criteria for the treatment and death databases that are used in the survival analysis.

The type of health insurance, region of residency, and treating hospital were assigned considering the most common value in the dataset. 32,560 patients had the same insurance over the study period, while 35,560 (97% of the dataset) had the same insurance for over 75% of the time. On the other hand, 35,942 patients never changed their region of residence over the study period (98% of the patients), and 35,217 patients were always treated in a hospital in their same region of residency. Only 5.4% of the patients received hospital care in a region different from their region of residency, at some point of their treatment.

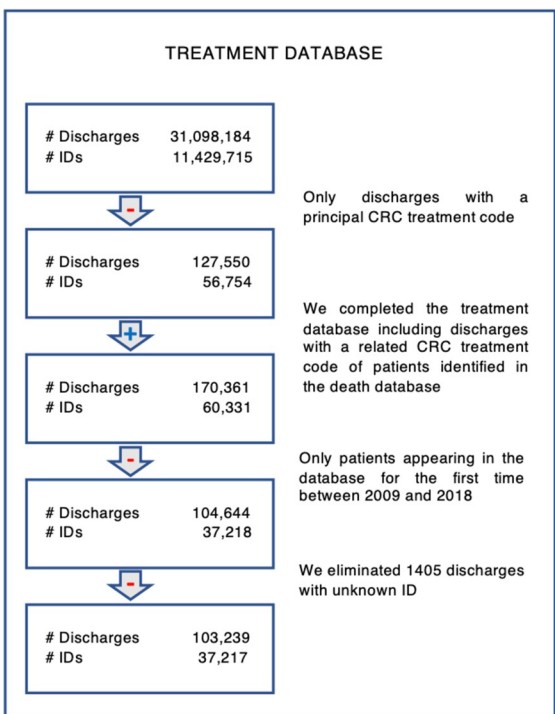
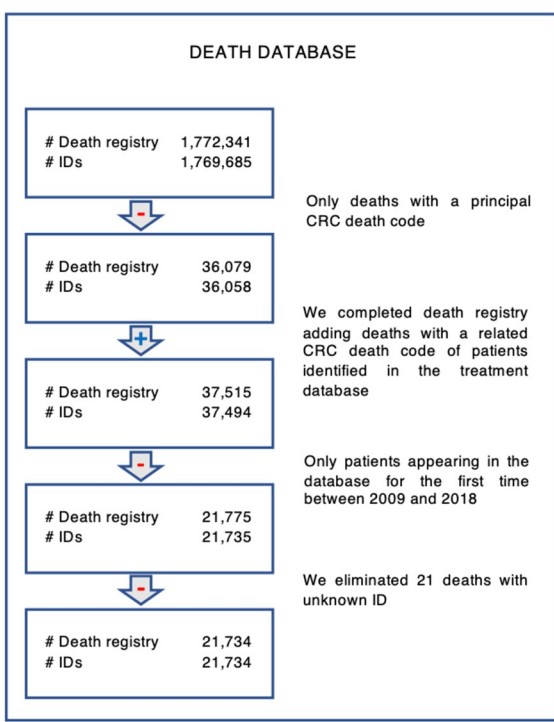

**Fig 1. Construction of treatment and death databases–inclusion and exclusion criteria.** Source: DEIS patient discharge database 2001–2019, DEIS mortality database 2000–2018.

**Table 1. Data description.** The first part of the table characterizes the treatment database, while the second part characterizes colorectal cancer deaths including those identified in the cohort and deaths from patients outside the cohort.

| | 2009 | 2010 | 2011 | 2012 | 2013 | 2014 | 2015 | 2016 | 2017 | 2018 | TOTAL |
|---|---|---|---|---|---|---|---|---|---|---|---|
| New identified CRC patients | 2,974 | 2,965 | 3,210 | 3,390 | 3,453 | 3,781 | 4,065 | 4,197 | 4,524 | 4,658 | 37,217 |
| Average Age (std) | 66.3 (14.0) | 66.6 (14.0) | 66.1 (14.1) | 65.9 (13.9) | 65.8 (14.0) | 66.2 (14.0) | 66.2 (13.6) | 66.2 (13.3) | 65.7 (13.6) | 66.1 (13.6) | 66.1 (13.8) |
| Women (%) | 51.3 | 51.6 | 52.2 | 51.3 | 50.2 | 49.4 | 50.5 | 50.0 | 48.9 | 49.0 | 50.3 |
| CRC deaths in cohort | 1,124 | 1,539 | 1,773 | 2,017 | 2,196 | 2,359 | 2,562 | 2,604 | 2,702 | 2,858 | 21,734 |
| Additional CRC deaths outside the cohort | 788 | 422 | 349 | 228 | 185 | 161 | 105 | 96 | 75 | 74 | 2,483 |
| Total CRC deaths | 1,912 | 1,961 | 2,122 | 2,245 | 2,381 | 2,520 | 2,667 | 2,700 | 2,777 | 2,932 | 24,217 |
| Average Age (std) | 71.5 (13.8) | 71.5 (13.8) | 71.6 (13.1) | 71.9 (13.2) | 70.9 (14.0) | 71.9 (13.5) | 71.7 (13.7) | 72.0 (13.6) | 71.6 (14.0) | 71.7 (13.7) | 71.6 (13.7) |
| Women (%) | 54.0 | 50.4 | 52.5 | 52.2 | 52.1 | 50.9 | 51.0 | 51.5 | 50.8 | 49.1 | 51.3 |

Table 1 presents the characteristics of the databases for the period under study, that are used to compute incidence, mortality, and prevalence rates. It is worth mentioning that for mortality we considered all deaths related to colorectal cancer, including patients outside the cohort, i.e., patients first identified before 2009 that died between 2009 and 2018 (a total of 2,483 deaths).

## 2.3 Methods

**2.3.1 Incidence, prevalence & mortality.** We computed total incidence, prevalence, and mortality rates (cases/100,000 inhabitants). For prevalence we included all CRC identified patients considering the complete DEIS patient discharge database 2001–2019.

Crude and age-standardized incidence and mortality rates were computed for the total population using the Segi World standard population table [18]. We also computed crude and age-standardized incidence and mortality rates for men and women, by type of insurance (ISAPRE and FONASA), and at regional level. We obtained the population by age segments by region and type of insurance from the National Socio-Economic Characterization Survey (Casen) [19].

**2.3.2 Survival analysis.** We performed empirical survival analysis and constructed a general survival model to consider simultaneously different observable covariates.

Since both databases share the anonymized patient ID codes, for each patient in the treatment database, we calculate their survival time as the total elapsed time between their first diagnosis and eventual appearance in the death database. If the cause of death corresponds to an unrelated ICD-10 code (neither principal nor related), then this is considered a right-censored case for our purposes. Similarly, if the patient does not appear in the death database, then they are considered surviving until the end of the database period (2018) with a right-censored death.

For the empirical survival analysis, we used the Kaplan–Meier estimator on different subsets of the patients' database, which correspond to relevant demographic subgroups, considering a confidence interval of 95%. We compared them using the log-rank test [20], for which we fixed a statistical significance level of 0.05.

We calculated the total survival rates, and explored potential differences, considering the type of health insurance, and the hospital type. We also studied the survival of patients with different health insurance (FONASA and ISAPRE), getting treatment in the same hospitals. For this purpose, we identified hospitals with a minimum of 100 patients, and at least a third

of them belonging to ISAPRE and another third or more to FONASA. It is worth noticing that in this category we only have private hospitals.

For each patient in the treatment database, we built a set of possible predictors for their survival. They include sex, health insurance, region of residency, age at diagnosis, year of diagnosis, type of hospital, and coverage by the GES guarantees.

The GES guarantees are determined by the year of treatment and the type of health insurance. CRC treatment was included in the GES plan in 2014, covering half of our research horizon. The GES program covers all the population with private and public insurance, except for those under the Military and Police Forces' health insurance. These patients were not affected by the introduction of the explicit guarantees of the GES program, which made them a natural control group for our survival analysis, allowing us to observe the impact of the program in the survival (if any) independently of any time-related trend.

Hospitals were classified into six different groups: private, belonging to the Police and Armed Forces, and four categories for public hospitals, according to their complexities defined by the Health Ministry (high complexity 1 for reference centers with a full portfolio of subspecialties, high complexity 2 with a set of subspecialties, and medium and low complexities). A map with the locations of the public hospitals appearing in the database is presented in (Fig 1 in S4 Appendix, right map).

Age of the patient at the time of diagnosis was grouped into five-year periods, allowing to explicitly incorporate nonlinear effects of the age covariate into the survival rates.

To build a general survival model we used the Cox proportional hazard model [21], treating categorical variables as dummies. We used the Akaike information criterion [22] to delineate the significant predictors from the variables mentioned above, but we forced the inclusion of the variables corresponding to the year of diagnosis and the coverage by the GES guarantees to identify the effect of GES by means of the control group, as mentioned above.

All statistical analyses were programmed using Python 3.8 with the *lifelines* package for survival analysis.

**2.3.3 Treatment of missing values.** The constructed death database has 21,734 patients, where 6,099 of them were not in the treatment database with a CRC principal or related hospital discharge between 2001 and 2018, i.e., they died of CRC without receiving treatment requiring hospitalization. Therefore, it is reasonable to assume that the stage of the cancer at the time of diagnosis was advanced enough not to recommend surgery or had comorbidities that prevented them from receiving more invasive treatments. This is also supported by the facts that i) most of these patients (72%) have other causes of hospitalizations, which indicates significant comorbidities that might have contributed to short survival periods after diagnosis, and ii) they are significantly older at death than those with CRC-related discharges (average of 76 versus 70 years old). See Table 5 in S3 Appendix for details.

Out of these 6,099 patients, 4,398 patients appear in the national registry of inpatient discharges due to hospitalizations for causes different from CRC, and therefore, using this information, we characterized them in terms of health insurance (which is not part of the national death registry). Furthermore, for the 1,701 remaining patients we performed a mean test analysis and determined that their characteristics such as age, sex, and cause of death are statistically the same as the first group of 4,398 patients. Therefore, by assigning a proportionally larger weight to the patients in the first group we performed the incidence and survival analyses correcting for the unknown health insurances.

For the 6,099 patients that died of CRC but did not have any CRC-related discharges, we considered them as diagnosed at the year of death. To further study the robustness of our estimates, we considered two additional scenarios: i) these patients were uniformly diagnosed in the previous 12 months before their deaths, and ii) these patients were diagnosed some time

before their deaths, consistently with the survival rates of stage IV CRC cancer patients. For this purpose, we used the survival rates reported in [23].

We had 21 unidentified patients in the death registry, representing less than 0.1% of the total cases, with no impact of our estimates. For incidence rates, we have 1,405 discharges with unknown ID. Considering each unidentified discharge as a new patient, they represented 3,8% of the total patients in the cohort. This is an upper bound since most of the identified patients appeared more than once in the discharge database (2.8 times on average). The yearly analysis of unknown discharges showed that 87% of the cases occurred during the first half of the study period, and that correcting by the average number of discharges in the database, the unknown patients would represent 2.5% of the total identified patients from 2009–2013 and only 0.4% for the period 2014–2018.

## 3. Results

### 3.1 Incidence, prevalence, and mortality

We computed crude and age-adjusted incidence and mortality rates to account for changes in the population (cases/100,000 inhabitants). Fig 2 presents a consistent increase in prevalence, and in crude and age-standardized incidence and mortality rates for the period under study.

Prevalence rate increased a 54%, while crude incidence and mortality rates increased in 38%. Age-standardize incidence and mortality rates increased in 16% and 13% respectively, from 2008 to 2019.

Table 2 presents crude and age-standardized incidence and mortality rates considering sex and principal types of health insurance, showing that all rates increase during the period under study.

The increase for men's crude incidence and mortality rates was larger than that for women's rates (46% and 52% vs. 31% and 37%, respectively). Similarly, for the age-standardized incidence and mortality, the increase in rates was larger for men than for women (19% and 22% for men vs. 4% and 12% for women).

We observe higher crude incidence and prevalence rates for FONASA patients, with a similar increase in the crude incidence rates (39% and 40% for FONASA and ISAPRE patients, respectively), but a markedly higher increase in the ISAPRE crude mortality rate (36% and 100% for FONASA and ISAPRE patients respectively) between 2009–2018. For age-

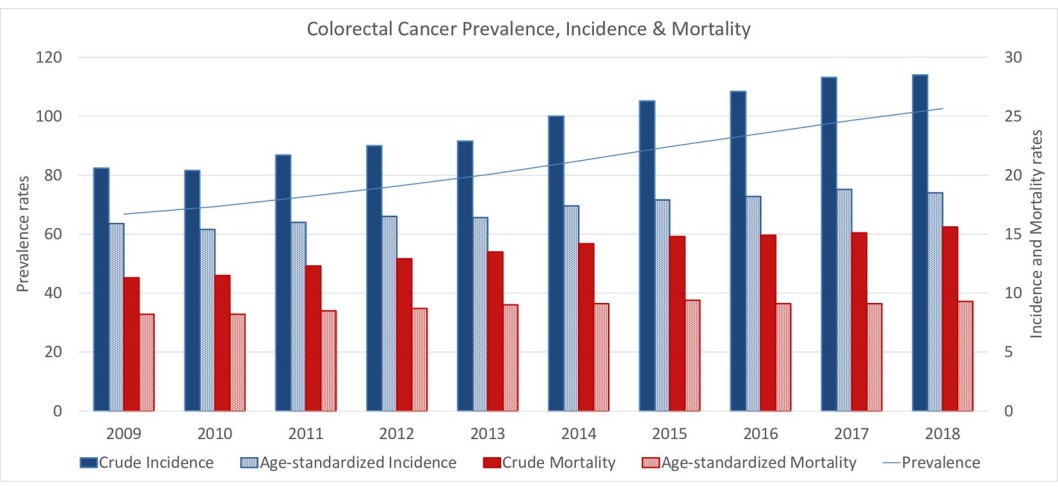

**Fig 2. Prevalence rate and crude and age-standardized incidence and mortality rates (cases/100,000 p.).**

**Table 2. Crude and age-standardized incidence and mortality rates, by sex (women/men), and health insurance (FONASA/ISAPRE).**

| | | | 2009 | 2010 | 2011 | 2012 | 2013 | 2014 | 2015 | 2016 | 2017 | 2018 |
|---|---|---|---|---|---|---|---|---|---|---|---|---|---|
| WOMEN | Crude | Incidence | 21.2 | 20.7 | 22.5 | 23.0 | 23.1 | 24.6 | 26.6 | 27.3 | 27.8 | 27.8 |
| | | Mortality | 12.0 | 11.4 | 12.7 | 13.2 | 13.9 | 14.2 | 14.9 | 15.1 | 15.1 | 15.2 |
| | Age Std | Incidence | 14.5 | 14.0 | 15.0 | 15.1 | 15.0 | 15.4 | 16.2 | 16.8 | 17.0 | 16.5 |
| | | Mortality | 7.6 | 7.1 | 7.6 | 7.9 | 8.1 | 7.9 | 8.1 | 8.0 | 8.1 | 7.9 |
| MEN | Crude | Incidence | 20.0 | 20.1 | 20.8 | 22.0 | 22.8 | 25.4 | 26.1 | 26.9 | 28.8 | 29.2 |
| | | Mortality | 10.6 | 11.6 | 11.9 | 12.5 | 13.2 | 14.1 | 14.8 | 14.6 | 15.1 | 16.1 |
| | Age Std | Incidence | 17.7 | 17.2 | 17.3 | 18.1 | 18.2 | 19.9 | 19.9 | 20.1 | 21.0 | 21.0 |
| | | Mortality | 9.1 | 9.7 | 9.7 | 9.9 | 10.2 | 10.6 | 10.9 | 10.5 | 10.5 | 11.1 |
| | | | 2009 | 2010 | 2011 | 2012 | 2013 | 2014 | 2015 | 2016 | 2017 | 2018 |
| FONASA | Crude | Incidence | 21.9 | 21.7 | 22.9 | 23.4 | 24.2 | 26.7 | 28.7 | 29.8 | 30.7 | 30.5 |
| | | Mortality | 12.2 | 12.1 | 12.6 | 13.7 | 13.9 | 15.0 | 16.0 | 15.9 | 16.3 | 16.6 |
| | Age Std | Incidence | 14.2 | 13.9 | 14.5 | 14.5 | 14.5 | 15.4 | 16.4 | 16.5 | 16.8 | 16.1 |
| | | Mortality | 7.5 | 7.3 | 7.4 | 7.7 | 7.8 | 8.0 | 8.3 | 8.0 | 8.0 | 8.0 |
| ISAPRE | Crude | Incidence | 16.1 | 15.1 | 14.9 | 17.2 | 15.5 | 18.4 | 18.9 | 18.4 | 21.1 | 22.7 |
| | | Mortality | 3.9 | 4.5 | 5.2 | 5.1 | 6.0 | 6.4 | 6.9 | 6.5 | 7.2 | 7.9 |
| | Age Std | Incidence | 17.8 | 16.6 | 15.7 | 18.8 | 16.2 | 19.5 | 20.0 | 17.8 | 18.7 | 19.0 |
| | | Mortality | 4.4 | 5.1 | 5.9 | 5.8 | 6.4 | 7.2 | 7.8 | 6.7 | 6.6 | 6.9 |

standardized rates, incidence of FONASA patients was lower than that of ISAPRE patients, while for mortality was the opposite, with higher age-standardize mortality rate for FONASA patients.

Table 3 shows crude and age-standardized incidence and mortality rates considering the geographical distribution, for the first and last year of the period under study. The regions are presented from north to south (a larger map of the country is presented in S4 Appendix).

**Table 3. Comparison of regional crude and age standardized incidence and mortality rates between the start and the end of the study period (yearly number of cases/100,000 p.).** The intensity of the color increases for higher rates. Map source: prepared by the authors from public domain polygonal data released by the Chilean National Library of Congress. A regional map is presented in S4 Appendix.

| | INCIDENCES RATES | | | | MORTALITY RATES | | | |
|---|---|---|---|---|---|---|---|---|
| | Crude | | Age Std | | Crude | | Age Std | |
| REGION | 2009 | 2018 | 2009 | 2018 | 2009 | 2018 | 2009 | 2018 |
| XV | 20.7 | 22.7 | 17.6 | 15.6 | 8.0 | 10.7 | 6.0 | 6.5 |
| I | 17.0 | 19.4 | 16.9 | 16.8 | 10.1 | 10.7 | 10.0 | 9.1 |
| II | 17.5 | 19.8 | 17.0 | 17.1 | 9.7 | 13.5 | 9.5 | 11.3 |
| III | 15.9 | 20.5 | 13.1 | 14.6 | 9.9 | 12.3 | 8.0 | 8.1 |
| IV | 18.5 | 27.6 | 14.0 | 17.6 | 11.8 | 18.1 | 8.8 | 10.6 |
| V | 24.7 | 33.2 | 16.5 | 19.2 | 14.9 | 18.6 | 9.3 | 9.7 |
| XIII | 21.4 | 28.1 | 16.9 | 19.1 | 10.9 | 13.8 | 8.1 | 8.5 |
| VI | 14.8 | 28.8 | 11.6 | 17.9 | 10.6 | 18.2 | 8.0 | 10.5 |
| VII | 17.6 | 28.8 | 13.4 | 17.6 | 11.1 | 13.8 | 7.7 | 7.9 |
| XVI | 23.4 | 27.4 | 15.8 | 15.5 | 12.0 | 18.4 | 7.5 | 9.5 |
| VIII | 19.5 | 31.6 | 15.3 | 20.0 | 10.9 | 18.8 | 8.2 | 10.8 |
| IX | 21.9 | 24.7 | 16.3 | 14.9 | 10.3 | 17.7 | 7.2 | 9.9 |
| X | 17.5 | 29.6 | 14.1 | 19.6 | 12.3 | 17.0 | 9.4 | 10.4 |
| XIV | 21.8 | 32.4 | 14.5 | 19.6 | 9.5 | 16.0 | 6.3 | 8.9 |
| XI | 21.1 | 31.1 | 19.5 | 23.0 | 13.1 | 17.0 | 11.4 | 11.5 |
| XII | 28.2 | 34.6 | 20.2 | 20.7 | 13.1 | 20.7 | 9.6 | 12.3 |

For crude incidence rates we observe an increase in all regions from 2009 to 2018. The rate increase varies from 10% in region XV (northern part of the country) to 95% in region VI (in the center). We also observe a geographical gradient, with higher incidence rates towards the south, that is most pronounced in 2018. On the other hand, the age-standardized incidence rates did not increase from 2009 to 2018 in all regions, with variations on the range of -11% in region XV (northern part of the country) to 39% in region X (southern part of the country). The geographical gradient is still present, and again, most pronounced in 2018.

For crude mortality rates we observe an increase in all regions, with a marked geographical gradient showing higher mortality rates towards the south, and which is most pronounced in 2018. The mortality rate increase varies from 6% in region I (northern part of the country) to 72% in regions VI, VIII and IX (in the center-south). Age standardized mortality rates increased in most regions and varied from -9% in region I (northern part of the country) to 41% I region XIV (southern part of the country). Age standardized mortality rates did not preserve a clear geographical gradient, however, in 2018, the lowest age-standardized mortality rate was 6.5 for the region XV in the extreme north and the highest was 12.3 for region XII in the extreme south.

We performed a robustness analysis with respect to the treatment of missing values, for both scenarios of the time of diagnosis: i) uniformly diagnosed in the previous 12 months before their deaths, and ii) diagnosed some time before their deaths, consistently with the survival rates of stage IV CRC cancer patients, as explained in subsection 2.3.3. For prevalence rate the maximum difference was only 4.5% in 2009 decreasing to 0% in 2018. For crude and age standardized incidence rates, the largest difference was observed in 2018 (6.7% and 5.4%, respectively) with an average difference during the study period of less than 2%. We notice that mortality rates were not affected since there were no ambiguities on death dates.

## 3.2 Survival analysis

**3.2.1 Kaplan–Meier analysis.** Analyzing the empirical Kaplan–Meier survival rate estimates, we observed an overall 43% five-year survival rate, considering all patients in the treatment database, which increased to 63% for the one-year survival rate. Fig 3 shows the five-year Kaplan–Meier survival curve for all patients in the treatment database and the 6,099 patients identified in the death registry. The Kaplan–Meier curves did not show significant differences between men and women in the empirical survival rates, and therefore, they are not included in the figures reported.

Remarkably, a significant difference was observed when considering the patients' health insurance, as shown in Fig 4 (p value < 0.05); the survival rates for FONASA groups C and D are not statistically different, but significantly differ from the survival outcomes of the other groups. Five-year survival rates were 64% for ISAPRE patients, 57% for Police and Armed Forces patients, 47% for FONASA D, 45% for FONASA C, 38% for FONASA B and 31% for FONASA A, showing a clear deterioration in the survival rates as the socioeconomic condition worsens.

To further explore potential differences in survival rates, we considered the type of hospital where patients got their CRC treatments. From Fig 5, we observed that hospitals could be classified in three major groups, according to their outcomes on patients' survival rates.

First, private and Police and Armed Forces hospitals, having a five-year survival rate of 58% and 55% survival rates, respectively. Second, public high complexity 1 and 2 hospitals have a five-year survival rate of 39% and 40%, respectively (we remark that these estimates are not statistically different at 95% confidence). Finally, the third group consists of medium and low complexity public hospitals, having a notoriously low five-years survival rate of 13%. For this group the 5-year survival rates are statistically the same.

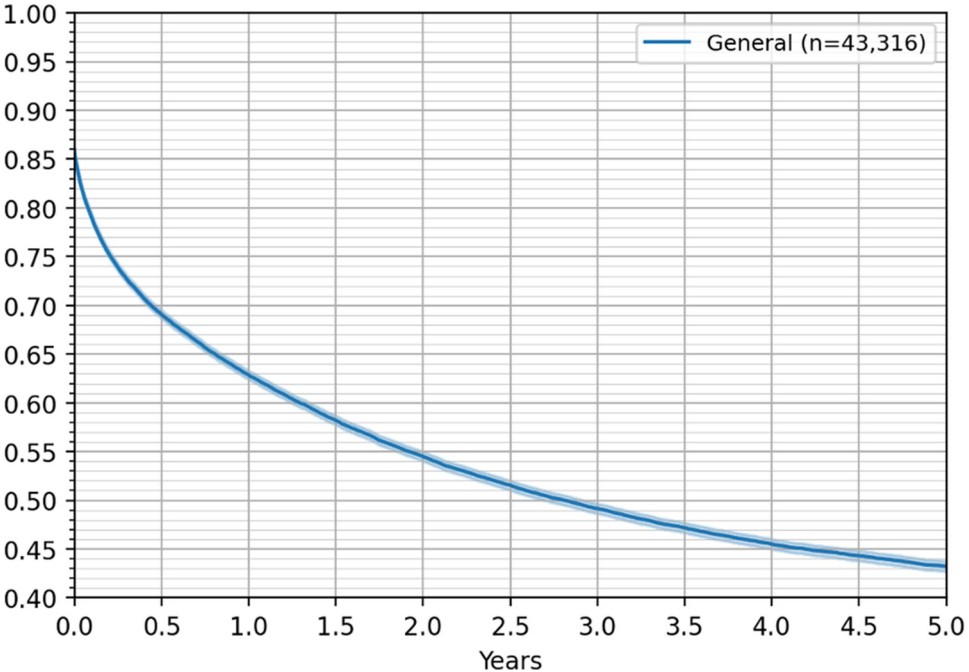

**Fig 3. Five-year Kaplan–Meier survival curve for all patients, with 95% confidence interval.**

Additionally, considering the subset of patients treated in hospitals with a large number of patients both from ISAPRE and FONASA, we observed significant differences in the five-year survival rates, with 57% for FONASA and 66% for ISAPRE patients, as shown in Fig 6.

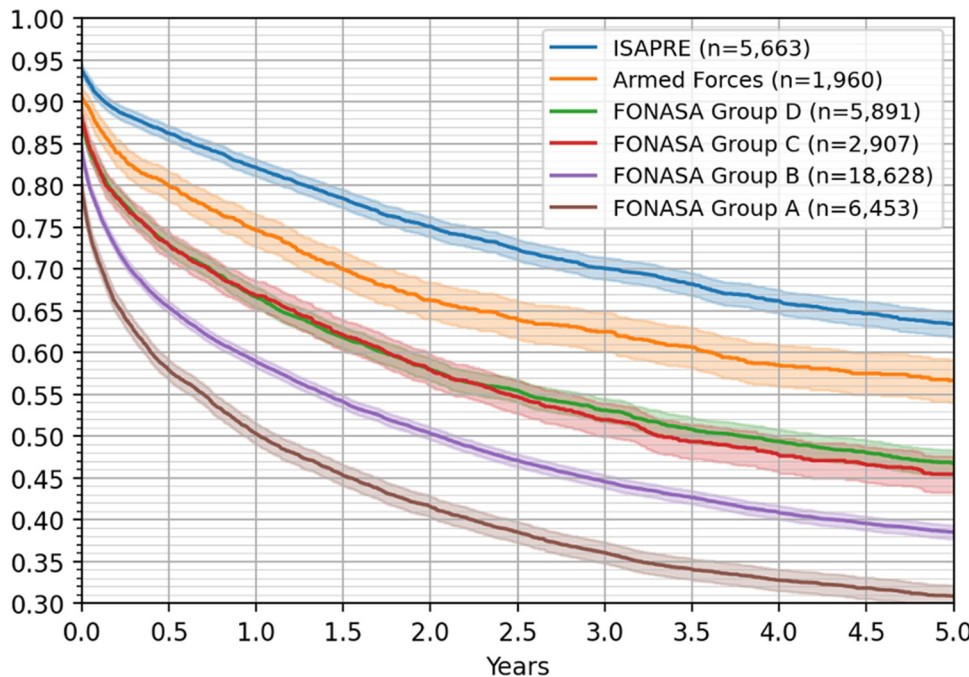

**Fig 4. Five-year Kaplan–Meier survival curves by health care insurance, with 95% confidence intervals.**

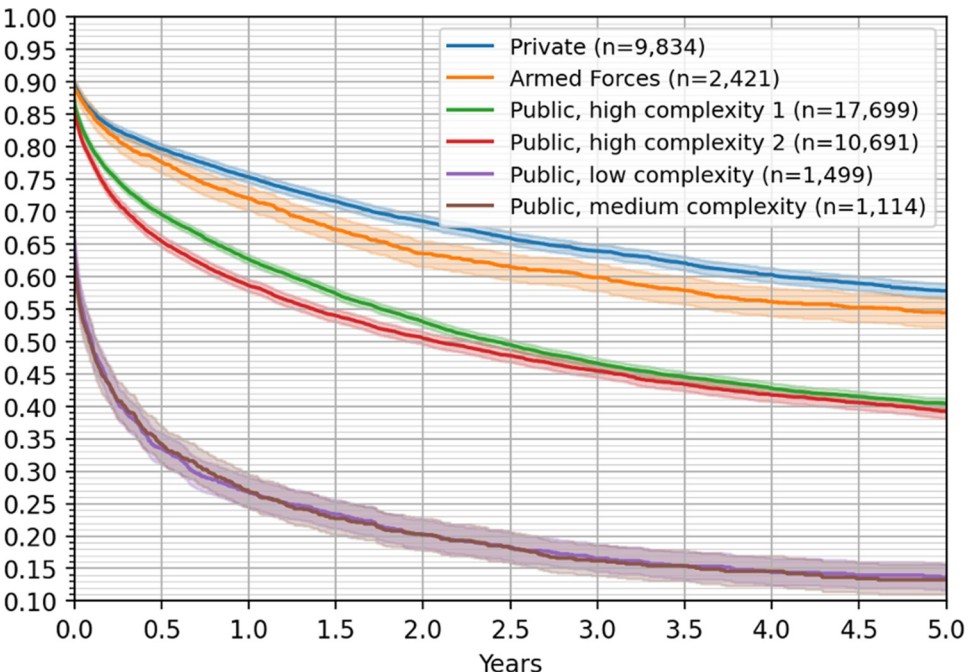

**Fig 5. Five-year Kaplan–Meier survival curves by type of hospital, with 95% confidence intervals.**

### 3.2.2 Cox proportional hazards model

We used the Cox proportional hazards regression model, or Cox model, to study the effect of different individual factors on the survival function. Factors considered include age, sex, year

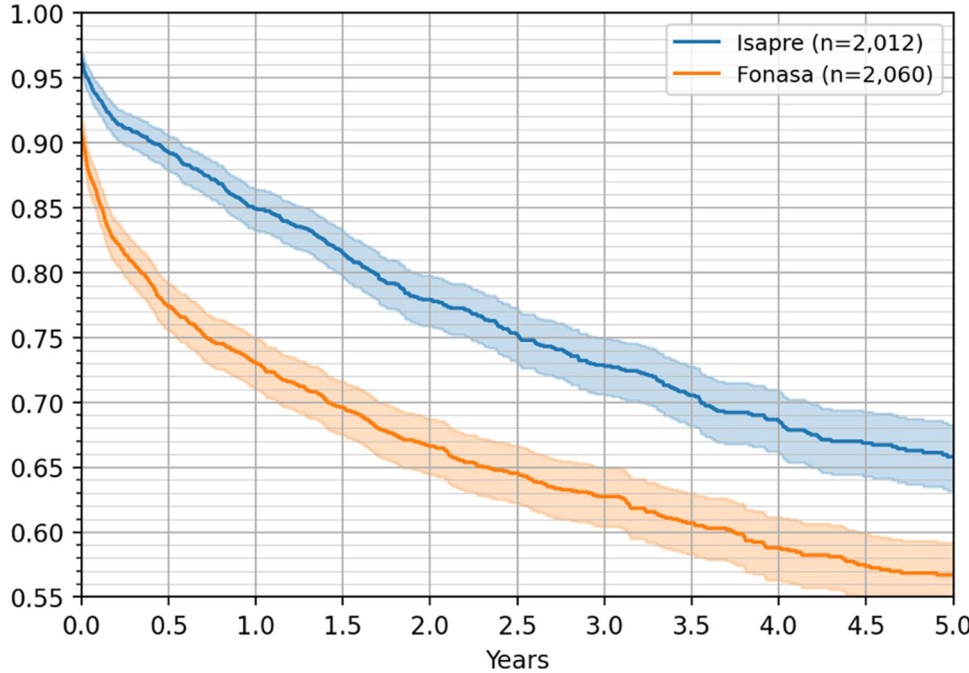

**Fig 6. Five-year Kaplan–Meier survival curves by insurance, for patients getting treatment in the same subset of hospitals, with 95% confidence intervals.**

of diagnosis, whether the patient was covered by GES since their CRC diagnosis, type of tumor (colon or rectal cancer), type of insurance, type of hospital, and geographical region.

We considered as the base case the group of female patients, 70–74 years old, under FONASA B insurance group, living in region XIII (where the capital of the country is located), with colon cancer, entering the treatment database during the GES period, and getting treatment in a high complexity 1 hospital. For this analysis, we grouped FONASA C and D patients. We also grouped some adjacent and statistically equivalent geographical regions to increase the number of patients in each group.

The results are summarized in Table 4, where the first column contains the coefficient with its associated standard deviation, the second column shows the corresponding hazard ratio at

**Table 4. Results for the Cox proportional hazards regression model.** Regions are ordered from north to south.

| | | Coefficient (std) | Hazard ratio | p value |
|---|---|---|---|---|
| AGE | 70–74 → 00–29 | -0.67 (0.10) | 0.51 | <0.005 |
| | 70–74 → 30–34 | -0.70 (0.10) | 0.50 | <0.005 |
| | 70–74 → 35–39 | -0.55 (0.07) | 0.58 | <0.005 |
| | 70–74 → 40–44 | -0.49 (0.06) | 0.61 | <0.005 |
| | 70–74 → 45–49 | -0.46 (0.05) | 0.63 | <0.005 |
| | 70–74 → 50–54 | -0.40 (0.04) | 0.67 | <0.005 |
| | 70–74 → 55–59 | -0.35 (0.03) | 0.70 | <0.005 |
| | 70–74 → 60–64 | -0.30 (0.03) | 0.74 | <0.005 |
| | 70–74 → 65–69 | -0.17 (0.03) | 0.84 | <0.005 |
| | 70–74 → 75–79 | 0.18 (0.03) | 1.19 | <0.005 |
| | 70–74 → 80–84 | 0.53 (0.03) | 1.70 | <0.005 |
| | 70–74 → 85+ | 1.00 (0.03) | 2.71 | <0.005 |
| | Female → Male | 0.09 (0.02) | 1.09 | <0.005 |
| | Year of diagnosis | -0.01 (0.01) | 0.99 | 0.06 |
| | With GES → No GES | 0.00 (0.03) | 1.00 | 0.98 |
| | Colon → Rectum | -0.11 (0.02) | 0.89 | <0.005 |
| INSURANCE | FONASA B → ISAPRE | -0.39 (0.04) | 0.68 | <0.005 |
| | FONASA B → Armed Forces | -0.16 (0.08) | 0.85 | 0.06 |
| | FONASA B → FONASA C+D | -0.06 (0.02) | 0.94 | 0.01 |
| | FONASA B → FONASA A | 0.37 (0.02) | 1.45 | <0.005 |
| HOSPITAL TYPE | High complexity 1 → Private | -0.28 (0.03) | 0.76 | <0.005 |
| | High complexity 1 → Armed Forces | -0.41 (0.08) | 0.66 | <0.005 |
| | High complexity 1 → High complexity 2 | 0.07 (0.02) | 1.08 | <0.005 |
| | High complexity 1 → Medium complexity | 0.79 (0.04) | 2.20 | <0.005 |
| | High complexity 1 → Low complexity | 0.71 (0.04) | 2.03 | <0.005 |
| REGION | XIII → I + XV | 0.12 (0.05) | 1.13 | 0.02 |
| | XIII → II | 0.30 (0.05) | 1.35 | <0.005 |
| | XIII → III + IV | 0.02 (0.04) | 1.02 | 0.52 |
| | XIII → V | -0.03 (0.03) | 0.97 | 0.29 |
| | XIII → VI | 0.23 (0.04) | 1.26 | <0.005 |
| | XIII → VII | 0.09 (0.04) | 1.09 | 0.02 |
| | XIII → XVI | 0.00 (0.04) | 1.00 | 0.93 |
| | XIII → VIII | 0.03 (0.03) | 1.03 | 0.26 |
| | XIII → IX | -0.07 (0.04) | 0.94 | 0.06 |
| | XIII → X | 0.10 (0.04) | 1.11 | 0.01 |
| | XIII → XIV | -0.14 (0.05) | 0.87 | <0.005 |
| | XIII → XI + XII | -0.11 (0.06) | 0.90 | 0.07 |

the coefficient's central value, and the third column shows the p value for the null hypothesis corresponding to equality of the base and the affected covariate.

Not surprisingly, the effect of age is statistically significant, and negatively correlated with the survival function (p value < 0.005). Moreover, the coefficients are ordered from smallest to largest (negative for younger than 70 and positive for older than 74). Sex also shows a significant impact, with men's surviving rates being lower than those of women's (p value < 0.005).

Year of diagnosis has a negative coefficient, indicating that patients diagnosed with CRC in the latter years of the study period had higher five-year survival rates than those diagnosed at the beginning of the period (p value 0.06). The GES program does not appear to have any effect on survival.

Having a tumor in the rectum is associated with an increase in survival probability compared with patients with colon tumors (p value < 0.005).

For the insurance type, when comparing the base case–FONASA B–we found that the survival for FONASA C+D is higher (p value 0.01) and for ISAPRE patients the survival rate is much higher (p value < 0.005). For patients in the Police and Armed Forces health insurance, we observe a survival rate that is between FONASA C+D and ISAPRE, but not statistically significant (p value 0.06).

The survival rates considering the type of hospital are statistically different (p value < 0.005) and higher for patients treated in private hospitals and those belonging to the Police and Armed Forces. However, for all other public hospitals (high complexity 2, medium, and low complexity), the patients' survival rates are lower than that for the patients treated in high complexity 1 hospitals, with significantly lower survival probability for patients treated in medium and low complexity hospitals.

Patients in the XIII region have, in general, a higher survival rate than patients in other parts of the country. The difference is statistically significant for regions I+XV and II (northern part of the country), regions VI and VII (center of the country), and region X (in the south). Region XIV (in the south of the country) is the only region with statistically higher survival rate compared to region XIII (p value <0.005).

The impact of some of the observable characteristics on the estimated survival probabilities is presented in Fig 7, where we compared the Cox hazard curves for several cases against the

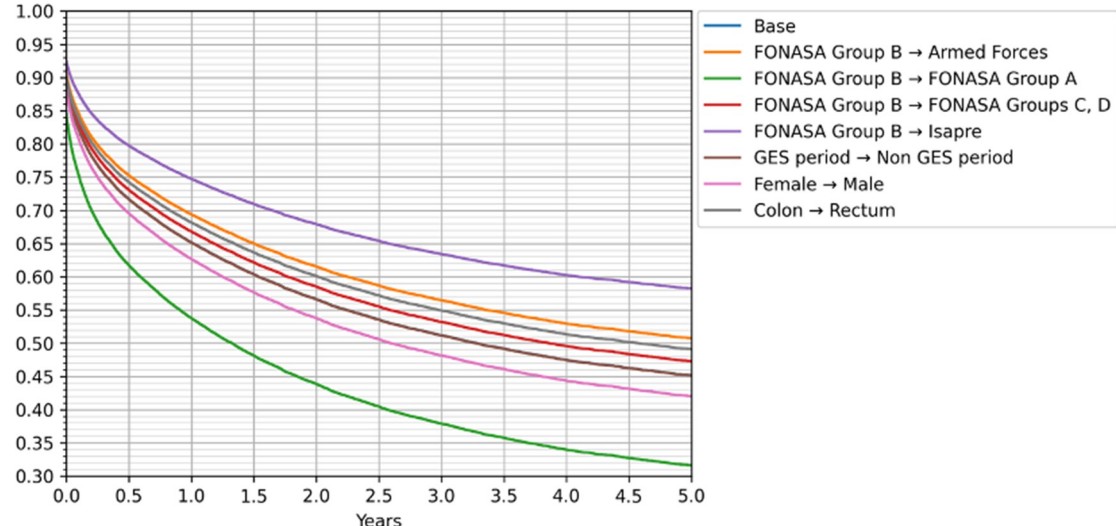

**Fig 7. Cox proportional hazards regression model.** Base: FONASA B, GES period, Female, Colon tumor, High complexity 1 public hospital, in the XIII Metropolitan region.

base case (female patient with colon tumor, diagnosed during the GES period, under FONASA B insurance and treated in a high complexity 1 hospital). In the figure, the base case is hidden by the non-GES period curve, showing the small impact the GES program has in our survival rate estimates.

## 4. Discussion

We have organized the discussion into three parts. In the first part, we discuss the main findings with respect the incidence, prevalence, mortality, and survival analysis. In the second part, we discuss the implications of our work for public policies, and in the last part we discuss the limitations of our study.

### 4.1 Main findings

**4.1.1 Incidence, prevalence & mortality.**   Previous national statistics estimated CRC incidence from the death registry and data on diagnosed cancer cases from 1998 to 2012 from four regional population registries [24]. In our study, we used individual-level information crossing hospital discharge and death registries to estimate incidence, prevalence, and mortality rates. This allows for more precise estimations without the need for extrapolations across time or regions.

Our results confirm a gradual, marked, and significant increase in CRC incidence (+38% for crude and +16% for age standardized), prevalence (+54%), and mortality (+38% for crude and +13% for age standardized) rates in the ten-year study period (2009–2018). It is noteworthy that the increase in the incidence and mortality was practically parallel in the study period, with only a slightly superior increase in age standardized incidence when compared to age standardized mortality. We found a *geographical gradient* with increasing incidence and mortality rates from the north to the south of the country.

The age-standardized incidence and mortality rates make it possible to compare Chile to other countries. CRC is strongly associated with a high human development index (HDI), probably related to sociodemographic, nutritional, and health care factors. Chile has one of the highest HDIs of the region, and thus, it is not surprising that the age-standardized incidence of CRC is also the highest compared to the rest of South America but significantly lower than European countries, Eastern Asia, Oceania and the United States [3, 25–27].

It is worth noticing that the values of crude incidence and mortality reflect the real number of CRC patients in Chile, thus they represent the true problem for health policies to confront.

**4.1.2 Survival analysis and its dependence from covariates.**   According to the American Cancer Society, the average 5-year survival rate of CRC is 65%, and the stage at diagnosis is one of the most determinant factors in survival: in localized disease (stage I), the 5-year survival is over 90% compared to 71% in regional disease, worsening to 14% in metastatic disease (stage IV) [28–31]. We found a 44% general 5-year survival rate, inferior to the abovementioned data from the USA. Unfortunately, we could not obtain information on disease stage from the Chilean public registries used. However, we explored the impact of other observable factors, such as age, region, health insurance, type of hospital, and the GES program, on survival rates.

From the Kaplan Meier analysis we found striking differences in survival rates when considering the health insurance of the patient and the type of hospital where the care was given. This social gradient has been reported in the literature from wealthy countries, with highlife expectancy and adequate health insurance coverage, demonstrating that a low socioeconomic position is a risk factor for lower survival, partially explained by a more advanced stage at diagnosis [32–35].

Sixty-three percent of patients affiliated with private health insurance (ISAPREs) were alive 5 years after CRC diagnosis, with a survival curve similar to that of developed countries [29, 30], which is significantly better than only 31% in the lowest income subgroup A of the public health insurance FONASA (see Fig 4).

The type of hospital also shows marked differences in the survival rates of their patients (see Fig 5). FONASA patients get treatment at their assigned public hospital, and patients insured by Police and Armed forces health system get care in their own hospitals. FONASA patients can be assigned to a private hospital under the GES program if the access guarantee cannot be met in a public facility. ISAPRE patients are, in most of the cases, treated in private facilities.

We observe that the lowest survival rates were found for patients treated in public low and medium complexity hospitals. A possible explanation for the lack of derivation to a high complexity facility is that the disease is advanced, and the health condition of the patient is poor, so further invasive and probably unsuccessful measures are not beneficial (see Fig 5). We also observed that even in high complexity facilities, public hospitals perform worse than private and Police and Armed Forces hospitals.

In FONASA, there are 920 patients per physician, which is significantly higher than the 276 patients per physician working in the private sector [36]. Additionally, ISAPRE patients have access to screening and preventive procedures included in most insurance plans, while FONASA patients can access such treatments only if there is a strong suspicion of CRC.

To further test the structural differences between FONASA and ISAPRE patients, we performed a survival analysis for patients treated in the same hospitals with different health insurances (see Fig 6). ISAPRE patients had a five-year survival rate that was 40% higher than those for FONASA patients, a difference that cannot be explained by the type of technology or quality of the physicians.

In the Cox survival model, the health insurance and the hospital type were statistically significant in explaining the survival probabilities, giving support to our exploratory Kaplan Meier analysis.

Age is clearly an important factor in survival. In our Cox analysis, all age groups were statistically different than the base case, with ordered coefficients with smaller hazard rates for younger patients, and a nonlinear effect (higher impact for older groups). Moreover, we performed a robustness analysis by including age as a linear term in the regression, resulting in a statistically significant variable with a positive coefficient, and even though the model specification had a worse fit, the main results were preserved.

In our model, we found that women survive longer than men, and that rectum patients also have a higher survival probability. Women performing better than men has been reported in the literature [37]. For the type of tumor, our model predicts slightly higher survival rates for rectum than colon tumors. However, coding problems are common and most of the studies consider all types of CRC together.

The coefficient for the year of diagnosis variable is negative, which implies a slight improvement in the survival probabilities in the later years, and can be explained by the introduction of new treatments and therapies.

In Chile, the GES program for CRC was implemented in 2014 to guarantee equity in prompt access to the best available treatment, with financial protection, once the patient is suspected to have CRC, independent of health care insurance. However, our study reveals that until 2018, survival rates had not significantly improved since the GES implementation. The GES program has proven to slightly decrease mortality rates in cervix, breast, and gallbladder cancer [38]. These diseases were included in the GES program in 2006 and either had a national screening program (Pap smear for women between 25 and 65 years old and mammography for women between 50 and 59 years old) or their prevention was part of the GES

program itself (cholecystectomy for people between 35 and 49 years old with gallbladder stones).

The question remains whether CRC results are not visible because five years is not enough time to assess the impact of the GES program or whether early diagnosis for CRC is the dominant factor that can only be managed with screening initiatives that are not currently included in the CRC GES program.

When considering the impact of the region on the patients' survival, we found better survival rate in the metropolitan region (XIII) than in the rest of the country. A plausible explanation in our statistics is known health access barriers, determined by the centralized distribution of resources, for example, the number of high-quality hospitals, a higher concentration of subspecialty physicians and access to prompt colonoscopy [36, 39]. In fact, the Metropolitan Region has a physician rate of 212 physicians/100,000 people, which is almost twice as high as that in the rest of the country (119 in and 117 physicians per 100,000 people in the northern and southern regions, respectively) [36].

## 4.2 Understanding the causes of CRC inequalities

In Subsection 4.1, we have discussed the inequalities observed in CRC survival rates among patients with different healthcare insurances, treating hospitals, and regions.

There are many paths that the healthcare authorities can follow to tackle these inequalities, such as the development and implementation of primary and secondary prevention programs, the provision of standardized (top of the line) treatments, the implementation of policies to improve the adherence to treatments and follow ups, among others. However, to develop and implement efficient and effective public policies, it is crucial to understand the reasons behind these inequalities, and thus to determine where to focus the efforts. We have classified the *possible* explanatory causes of these inequalities in four groups:

a) Patients with private and public healthcare insurances have different health status, prior to CRC. Thus, patients with public healthcare plans, which belong to lower socioeconomic groups, have a worse general health condition, presenting more comorbidities, poorer nutritional habits, less physical activities, among others. In this case, primary prevention programs aim to develop a healthy lifestyle, with balanced nutrition and regular physical activity, obesity prevention, and tobacco and alcohol use prevention would be recommended. In Chile, there is a nationwide program promoting healthy lifestyles (www.eligevivirsano.cl), but with no specific focus on preventing CRC, and a very limited budget.

b) Patients with public health insurance have significantly less access to screening tests, and are most likely diagnosed at later stages of the disease. Early diagnosis is only possible with screening programs. There is evidence that there is a need for long-standing screening programs to see the impact on CRC reduction [3, 40]. In Chile, we have had some localized CRC screening programs with promising results on early CRC diagnosis [41, 42]. However, there is no national CRC screening program and, therefore, patients have access to screenings arbitrarily based on doctors' recommendations, personal requirements, and economic resources.

c) Although the GES program guarantees the treatment for all people 15 years and older with suspected and diagnosed CRC, it is unclear if all patients receive the exact same treatment. The GES program guarantees a set of procedures and medications that is not necessarily updated on a regular basis, and therefore, there is a delay in incorporating new drugs or treatments. However, patients under the private health insurance have complementary coverage for high-cost treatment having access to newer drugs that could make a difference, especially for patients at advanced CRC stages. Moreover, even though the GES plan has opportunity guarantees (maximum times for patients to be treated), waiting lists for FONASA patients are

not public, and there is evidence that due dates are not always met, resulting in delays in diagnosis and treatment [43].

d) Finally, once diagnosed and under treatment, CRC patients in both groups might have different percentage of adherence to treatments. Multiple reasons might explain this difference, such as education, family, and work environment, among others.

To understand the reasons that explain these inequalities is part of our ongoing research.

### 4.3 Study strengths and limitations

To our knowledge, this is the first study to compare and relate information from different registries. Our results provide objective data about the steady increase in incidence, prevalence and mortality of CRC and situates Chile in the highest range of South America, approaching developed countries. In addition, to our knowledge, lethality and survival were analyzed for the first time at the national level, demonstrating a clear dependence on regional and socioeconomic factors.

Our study has several limitations. First, our results depend on the limitations and correctness of national databases. Second, we could identify only patients with hospital admissions or those who died from CRC without hospital admission but not people who had only ambulatory colonoscopic resection of a lesion with "cancer in situ". Third, we do not know the stage of the disease at the time of diagnosis, a factor of utmost prognostic importance.

Therefore, as part of our ongoing research, we are developing mathematical and econometric models to estimate the distribution of CRC diagnoses among the different stages at the time of diagnosis. Furthermore, we also quantified the effect of different screening programs on the cancer stages at diagnosis as a potential explanation for the difference in survival rates observed between private and publicly insured patients.

## 5. Conclusions

Our data and results show the real situation of CRC and its changes across the country in the period 2009–2018. Reliable individual patient information was obtained, allowing us to estimate the incidence and mortality of colorectal cancer in Chile. The factors influencing the survival of patients were analyzed. Our study provides evidence for the impact of socioeconomic inequalities on CRC incidence, mortality, and 5-year survival rates. Incidence and mortality rates were considerably lower, and the survival was significantly longer for patients enrolled in the private insurance system than for those enrolled in the public national health insurance system. Even within the publicly insured group, clear differences were observed in survival depending on socioeconomic conditions and treating hospital. Moreover, we could not find a significant effect of the GES program in improving the survival of CRC patients.

Our work provides valuable information that helps to compare Chile with the rest of the world. It provides a solid basis for the construction of new health care programs with differential emphasis in different regions, which can truly improve the reality of CRC.

## Supporting information

**S1 Appendix. FONASA groups.**
(DOCX)

**S2 Appendix. CRC ICD-10 diagnosis codes.**
(DOCX)

**S3 Appendix. ICD-10 discharge codes for CRC-deceased patients without CRC-related discharges.**
(DOCX)

**S4 Appendix. Geography of Chile: Regions and public hospitals.**
(DOCX)

## Author Contributions

**Conceptualization:** Susana Mondschein, Felipe Subiabre, Natalia Yankovic, Camila Estay, Christian Von Mühlenbrock, Zoltan Berger.

**Data curation:** Susana Mondschein, Felipe Subiabre, Natalia Yankovic.

**Formal analysis:** Susana Mondschein, Felipe Subiabre, Natalia Yankovic, Camila Estay, Christian Von Mühlenbrock, Zoltan Berger.

**Investigation:** Susana Mondschein, Felipe Subiabre, Natalia Yankovic, Camila Estay, Christian Von Mühlenbrock, Zoltan Berger.

**Methodology:** Susana Mondschein, Felipe Subiabre, Natalia Yankovic, Camila Estay, Zoltan Berger.

**Project administration:** Susana Mondschein.

**Resources:** Susana Mondschein.

**Software:** Susana Mondschein, Felipe Subiabre.

**Supervision:** Susana Mondschein, Natalia Yankovic, Zoltan Berger.

**Validation:** Susana Mondschein, Felipe Subiabre, Natalia Yankovic, Zoltan Berger.

**Visualization:** Susana Mondschein, Felipe Subiabre.

**Writing – original draft:** Susana Mondschein, Felipe Subiabre, Natalia Yankovic, Camila Estay, Christian Von Mühlenbrock, Zoltan Berger.

**Writing – review & editing:** Susana Mondschein, Natalia Yankovic, Camila Estay, Zoltan Berger.

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
