## [Decision Letter · Decision Letter 0]

27 Apr 2022

PONE-D-22-00874Colorectal cancer trends in Chile: a Latin-American country with marked socioeconomic inequitiesPLOS ONE

Dear Dr. Mondschein,

Thank you for submitting your manuscript to PLOS ONE. After careful consideration, we feel that it has merit but does not fully meet PLOS ONE’s publication criteria as it currently stands. Therefore, we invite you to submit a revised version of the manuscript that addresses the points raised during the review process.

We look forward to receiving your revised manuscript.

Kind regards,

Keith Anthony Dookeran, MD PhD

Academic Editor

PLOS ONE

Journal Requirements:

2.Thank you for stating the following financial disclosure: 

(No)

3. We noted in your submission details that a portion of your manuscript may have been presented or published elsewhere. 

(It was initialle submitted to Plos Medicine (PMEDICINE-D-21-05258) and it was internally transfered to Plos One as suggested by the editor of Plos Medicine)

5. Please update your submission to use the PLOS LaTeX template. The template and more information on our requirements for LaTeX submissions can be found at http://journals.plos.org/plosone/s/latex.

6. Please ensure that you refer to Figures 10 and 11 in your text as, if accepted, production will need this reference to link the reader to the figure.

7. We note that Figure 4 in your submission contain map images which may be copyrighted. All PLOS content is published under the Creative Commons Attribution License (CC BY 4.0), which means that the manuscript, images, and Supporting Information files will be freely available online, and any third party is permitted to access, download, copy, distribute, and use these materials in any way, even commercially, with proper attribution. For these reasons, we cannot publish previously copyrighted maps or satellite images created using proprietary data, such as Google software (Google Maps, Street View, and Earth). For more information, see our copyright guidelines: http://journals.plos.org/plosone/s/licenses-and-copyright.

a. You may seek permission from the original copyright holder of Figure 4 to publish the content specifically under the CC BY 4.0 license.  

8. We note you have included a table to which you do not refer in the text of your manuscript. Please ensure that you refer to Tables 3, 4, 5, 6, 7 in your text; if accepted, production will need this reference to link the reader to the Table.

Reviewers' comments:

Reviewer's Responses to Questions

**Comments to the Author**

1. Is the manuscript technically sound, and do the data support the conclusions?

Reviewer #1: No

Reviewer #2: Partly

2. Has the statistical analysis been performed appropriately and rigorously? 

Reviewer #1: No

Reviewer #2: Yes

3. Have the authors made all data underlying the findings in their manuscript fully available?

Reviewer #1: Yes

Reviewer #2: Yes

4. Is the manuscript presented in an intelligible fashion and written in standard English?

Reviewer #1: Yes

Reviewer #2: Yes

5. Review Comments to the Author

Reviewer #1: In this study Mondschein, et al. describe trends in colorectal cancer (CRC) incidence, survival, and mortality in Chile. While other studies on CRC in Chile have been published, this appears to be the first paper (or at least the first in some time) using national-level registry data. Unfortunately, in my opinion the paper’s introduction and objectives lack a clear focus, which translates into a poorly organized results section. This, coupled with a number of methodological problems means I cannot recommend this paper for publication.

Major points:

BMI/obesity is one of, if not the largest, risk factor for CRC and it also plays a very important role in screening for it. However, it is not mentioned once in the entire study. If BMI data are not available from the source registries, then this represents a severe methodological problem with the study as the authors will not be able to answer how much of the trends in CRC are driven by/associated with similar trends in BMI/obesity.

Roughly 1/3 of individuals from each arm of the study were excluded due to unknown/unavailable IDs, but the authors do not specify what exactly this means. At a minimum, the characteristics of these individuals must be compared to those retained in the study to ensure that there is little to no bias in the analysis.

Why are the main figures presenting crude rates but the age-standardized rates are relegated to the appendix? Age-standardized would be better.

Several times in the results rates are compared between groups (e.g., men vs. women, type of insurance). However, there are no statistics to back up these comparisons, nor any 95% confidence intervals to convey the reliability of the estimates. While the overall sample size is large enough that any such comparisons are likely to be statistically significant, the authors cannot draw conclusions based solely on qualitative comparisons between subgroups.

I think that the authors’ repeated emphasis of age-stratified results is also problematic. It’s not part of their main objective, to my knowledge there isn’t any reason to think that age-related trends in cancer are different in Chile than elsewhere, and it detracts from the focus on the more interesting/unique results presented (e.g., regional). This focus isn’t justified until the first part of the discussion, but because the increase in incidence from 2009 to 2018 isn’t presented in a way that allows the reader to compare this increase across different age groups I have no way to verify the authors’ statement of a marked incidence increase being observed in patients age 35-50. This age stratum is not presented in any of their tables or figures- how are we to know where it came from?

Categorizing age into five-year increments for purposes of multivariable modeling is not well-justified by the study data or objectives. In general these variables are best kept in their continuous form as it maximizes the information incorporated into the model. Additionally, all of the point estimates for the various age categories take up space and make it harder to follow the main message of the table. This is further complicated by the authors’ decision to use a category in the middle of their distribution as the referent.

The results section in general suffers from poor organization. Most paragraphs are presented in a different order than their corresponding tables and figures (for example, the discussion of crude mortality rates requires the reader to turn back two pages to refer to Figure 2). Many paragraphs refer to supplementary material, but some of these tables are rather dense and the text gives few clues to help the reader navigate them. Parts of the results text also belong in the methods.

Region numbers in Table 2 are presented without a definition or any context- they aren’t even labeled on the map. How is the reader supposed to interpret them? Are these states or provinces as defined by the Chilean government, or did the authors use some other definition?

In the discussion, the authors refer to the increase in incidence, prevalence, and mortality but it is unclear whether this refers to crude or age-standardized rates.

The discussion also highlights the role that resource availability plays in specific insurance plans- this is a serious limitation to the authors’ attempt to link insurance to socioeconomic status when the observed effects could just as easily reflect access to care (or access to high-quality care). I realize that this is probably a limitation of the registry datasets, but even so I would be more comfortable if this paper referred to it as insurance status or access to health care, and not socioeconomic status. This analysis will not be comparable to other studies of social determinants as its groups were defined in ways highly specific to the Chilean population.

Minor points:

Table 1 would function better as a figure (e.g., line plot) to enable the reader to better evaluate the trends over time.

The results text (e.g., last paragraph in section 3.1) do not make it clear that crude prevalence and mortality are also presented in the figure (though the caption does make this clear).

Reporting numeric results to two decimal places is standard- Table 2 is particularly hard to follow due to all the decimal places present in it.

Reviewer #2: Colorectal cancer trends in Chile: a Latin-American country with marked socioeconomic inequities

Overall

This paper provides important insight into the status of prevalence, incidence, and mortality rates related to colorectal cancer. The authors’ methods seem sound despite the common limitations related to using claims data for this type of research. They use a novel approach in describing the colorectal cancer environment using individual data rather than relying on population level data. My primary concerns regarding the manuscript are described in detail below. Of note, I am not familiar with the health care system in Chile, so please take careful consideration of any comments that may be attributable to my misconceptions of the Chilean health care system, which are solely based on what is presented in the manuscript.

Primary concerns

1. In the abstract, the authors state that “a national screening program with rapid access to diagnostic and therapeutic procedures is the only way to diminish serious inequality”. This is the strong statement and although these would likely lead to improvements in the observed CRC rates, this statement is not support by evidence in the manuscript.

2. The author state that there were 6,626 patients included in the death database and not the treatment database, did the authors consider doing a sensitivity analysis of why those patients were not captured in the treatment database and how their characteristics compared to those who were included in the analyses?

3. It is unclear whether covariates were assigned based on first entry into the treatment registry and how changes in insurance status or geographic region were captured over time and how they were addressed in the modeling. Additional description of these methods is recommended.

4. It is unclear if there is missing data on covariates in the two databases (other than exclusion of unknown sex as described in the methods section) and how it was handled during analyses. The authors should include a description of the distribution of missing data and corresponding methods or provide a statement if there was no missing data. Furthermore, I would recommend further discussion on who may not be captured in the treatment database and how that might affect the reporting of their results

5. The authors use 2014 as a starting point for the effects of the GES program. I do not know how the program was implemented, but did the authors consider using a lag period to assess the impact of the GES program as it’s effects might not have had immediate effects during the early implementation period?

6. The authors reference different regions within Chile, however these regions are not noted on any of the included maps. I would recommend that a map be included that note the region numbers for those who are not familiar with the geography of Chile.

7. The authors state differences in incidence and mortality rates based on FONASA public insurance versus privately insured ISAPRES. They use insurance as a proxy for individual socioeconomic position, but do not discuss potential structural or institutional factors related to these insurance practices that may be driving the observed results (i.e., institutional resources, etc.). This is partially discussed in reference to geography, but not insurance status.

8. In reference to geographic differences in CRC rates: Since the findings are based on hospitalization records and not patient residence, how might the observed rates be attributed to population migration or whether individuals travel to certain regions for advanced medical care.

9. Cox proportional hazard model results are reported as odds ratios. These should be expressed in hazard ratios.

10. In the discussion, the authors state that the observed geographic gradient parallel ethnic differences in the country. Are the authors implying genetic differences? Socioeconomic factors? Geographical factors? I think the authors should further explain the potential ethic gradient to ensure that undue “blame” is not being placed on potentially minoritized populations when social or structural factors may be primary drivers.

Minor typos

1. Page 10, line 1: “familiar economic consequences…” should be familial.

2. Figure 2: “Colorrectal” is misspelled. Should be “Colorectal”. R-squared value for mortality only has 3 significant figures compared to the four significant figures for the other rates.

6. PLOS authors have the option to publish the peer review history of their article (what does this mean?). If published, this will include your full peer review and any attached files.

Reviewer #1: No

Reviewer #2: No

---

## [Author Response · Author response to Decision Letter 0]

7 Jun 2022

Dear Sirs,

We thank you for giving us the opportunity to submit a revised draft of our article titled “Colorectal cancer trends in Chile: a Latin-American country with marked socioeconomic inequities”, we appreciate the time and effort that you and the reviewers have dedicated to providing your valuable feedback. We have been able to incorporate changes to reflect most of the suggestions provided by the reviewers. We have highlighted the changes within the manuscript. Here is a point-by-point response to the reviewers’ comments and concerns.

REVIEWER #1: 

In this study Mondschein, et al. describe trends in colorectal cancer (CRC) incidence, survival, and mortality in Chile. While other studies on CRC in Chile have been published, this appears to be the first paper (or at least the first in some time) using national-level registry data. Unfortunately, in my opinion the paper’s introduction and objectives lack a clear focus, which translates into a poorly organized results section. This, coupled with a number of methodological problems means I cannot recommend this paper for publication.

We appreciate very much your feedback which allowed us to improve the new version of our paper. Following your advice, we have reorganized and focused the contents of each section, having now consistency among all of them. We have narrowed the objective of our study to show the significant inequalities in CRC outcomes that are still present in Chile, despite the efforts that have been implemented in recent year, such as the universal GES plan. We now concluded our discussion with possible explanations for this disparity, which are the focus of our ongoing research.

Major points:

• BMI/obesity is one of, if not the largest, risk factor for CRC and it also plays a very important role in screening for it. However, it is not mentioned once in the entire study. If BMI data are not available from the source registries, then this represents a severe methodological problem with the study as the authors will not be able to answer how much of the trends in CRC are driven by/associated with similar trends in BMI/obesity. 

We fully agree with you that one of the potential causes of the increase in CRC incidence is obesity (or BMI), and furthermore, other risks factors associated to changes in lifestyle. The national registry of hospitals discharges does not have such covariates, this database reports sociodemographic characteristics and treatments. However, as it is mentioned in the revised manuscript, the focus of our study is to identify if inequalities in terms of health outcomes are still present despite the introduction of the GES plan. For this purpose, we study differences in terms of gender, health insurance plans, treating hospitals, among others. We fully agree that the next step is to understand the root causes of such disparities. For example, patients with public health insurance might have more comorbidities, including higher percentages of obesity which might partly explain lower survival rates. We included this analysis in the discussion and highlight that this is part of our ongoing research (see Discussion section).

• Roughly 1/3 of individuals from each arm of the study were excluded due to unknown/unavailable IDs, but the authors do not specify what exactly this means. At a minimum, the characteristics of these individuals must be compared to those retained in the study to ensure that there is little to no bias in the analysis.

Thank you for your comment. We fully agree that further explanations were needed. Therefore, in Section 2, we have included a detailed analysis regarding missing or unavailable data, and how it was incorporated in the study.

“The constructed death database has 21,734 patients, where 6,099 of them were not in the treatment database with a CRC principal or related hospital discharge between 2001 and 2018, i.e., they died of CRC without receiving treatment requiring hospitalization. Therefore, it is reasonable to assume that the stage of the cancer at the time of diagnosis was advanced enough not to recommend surgery or had comorbidities that prevented them from receiving more invasive treatments. This is also supported by the facts that i) most of these patients (72%) have other causes of hospitalizations, which indicates significant comorbidities that might have contributed to short survival periods after diagnosis, and ii) they are significantly older at death than those with CRC-related discharges (average of 76 versus 70 years old). See Appendix C for details.

Out of these 6,099 patients, 4,398 patients appear in the national registry of inpatient discharges due to hospitalizations for causes different from CRC, and therefore, using this information, we characterized them in terms of health insurance (which is not part of the national death registry). Furthermore, for the 1,701 remaining patients we performed a mean test analysis and determined that their characteristics such as age, sex, and cause of death are statistically the same as the first group of 4,398 patients. Therefore, by assigning a proportionally larger weight to the patients in the first group we performed the incidence and survival analyses correcting for the unknown health insurances.

For the 6,099 patients that died of CRC but did not have any CRC-related discharges, we considered them as diagnosed at the year of death. To further study the robustness of our estimates, we considered two additional scenarios: i) these patients were uniformly diagnosed in the previous 12 months before their deaths, and ii) these patients were diagnosed some time before their deaths, consistently with the survival rates of stage IV CRC cancer patients. For this purpose, we used the survival rates reported in [Arhi et al., Stage at diagnosis and survival of colorectal cancer with or without underlying inflammatory bowel disease: a population-based study].

We had 21 unidentified patients in the death registry, representing less than 0.1% of the total cases, with no impact of our estimates. For incidence rates, we have 1,405 discharges with unknown ID. Considering each unidentified discharge as a new patient that would represent 3.8% of the total patients in the cohort. This is an upper bound since most of the identified patients appeared more than once in the discharge database (2.8 times on average). The yearly analysis of unknown discharges showed that 87% of the cases occurred during the first half of the study period, and that correcting by the average number of discharges in the database, the unknown patients would represent 2.5% of the total identified patients from 2009-2013 and only 0.4% for the period 2014-2018.”

• Why are the main figures presenting crude rates, but the age-standardized rates are relegated to the appendix? Age-standardized would be better.

Following your suggestion, we have included all age-standardized rates in the main text. We agree that age-standardized rates are more insightful when comparing with the reality worldwide. We also included the crude rates, that are useful when designing and implementing public policies to tackle the issues discussed in the paper.

• Several times in the results rates are compared between groups (e.g., men vs. women, type of insurance). However, there are no statistics to back up these comparisons, nor any 95% confidence intervals to convey the reliability of the estimates. While the overall sample size is large enough that any such comparisons are likely to be statistically significant, the authors cannot draw conclusions based solely on qualitative comparisons between subgroups. 

Thank you for your comment. We have made improvements in the results to make sure conclusions are backed up with solid statistical analyses. All Kaplan-Meier analyses include 95% confidence intervals, and for the Cox model we report the coefficients, their standard deviations, hazard rates and the p values. 

• I think that the authors’ repeated emphasis of age-stratified results is also problematic. It’s not part of their main objective, to my knowledge there isn’t any reason to think that age-related trends in cancer are different in Chile than elsewhere, and it detracts from the focus on the more interesting/unique results presented (e.g., regional). This focus isn’t justified until the first part of the discussion, but because the increase in incidence from 2009 to 2018 isn’t presented in a way that allows the reader to compare this increase across different age groups, I have no way to verify the authors’ statement of a marked incidence increase being observed in patients age 35-50. This age stratum is not presented in any of their tables or figures- how are we to know where it came from?

Thank you for your comment, we fully agree with you that this analysis is not the focus of our paper. Following your suggestion, we have reorganized the results and enhanced the comparison between regions including the type of treating hospital into the analysis. We fully agree that having CRC survivals that depend on the age (and even on the sex) of patients is not a novel result and not as interesting as the inequalities found considering health insurance, regions, and hospital types.

• Categorizing age into five-year increments for purposes of multivariable modeling is not well-justified by the study data or objectives. In general, these variables are best kept in their continuous form as it maximizes the information incorporated into the model. Additionally, all of the point estimates for the various age categories take up space and make it harder to follow the main message of the table. This is further complicated by the authors’ decision to use a category in the middle of their distribution as the referent.

We use five-years intervals to allow for a nonlinear impact as age increases. For checking robustness of this election, we have performed the analysis using alternative model specifications (with polynomial models on continuous age of degree one, two and three, to improve the fit) and the conclusions did not change. As it was mentioned before, we are now giving less relevance to the impact of age into survival rates and focusing on the observed inequalities (health insurance, type of hospital, and region). This discussion is now included in the Methods section of the revised manuscript.

• The results section in general suffers from poor organization. Most paragraphs are presented in a different order than their corresponding tables and figures (for example, the discussion of crude mortality rates requires the reader to turn back two pages to refer to Figure 2). Many paragraphs refer to supplementary material, but some of these tables are rather dense and the text gives few clues to help the reader navigate them. Parts of the results text also belong in the methods.

We fully agree with your comment. Following your suggestion, we have reorganized the paper and simplified the information that is presented. We are now keeping the same structure from methods, results, and discussion sections helping readers to navigate through the text.

• Region numbers in Table 2 are presented without a definition or any context- they aren’t even labeled on the map. How is the reader supposed to interpret them? Are these states or provinces as defined by the Chilean government, or did the authors use some other definition?

Thank you for noticing the lack of context in presenting the results. We are presenting the regions from north to south, using the coding system of the Chilean government. We are including a geopolitical map of the country (identifying the regions) to show the challenges of such a unique geography. 

• In the discussion, the authors refer to the increase in incidence, prevalence, and mortality but it is unclear whether this refers to crude or age-standardized rates.

Following your suggestion, we are now mentioning crude and age-standardized incidence and mortality rates. Prevalence is always crude to simplify its interpretation.

• The discussion also highlights the role that resource availability plays in specific insurance plans- this is a serious limitation to the authors’ attempt to link insurance to socioeconomic status when the observed effects could just as easily reflect access to care (or access to high-quality care). I realize that this is probably a limitation of the registry datasets, but even so I would be more comfortable if this paper referred to it as insurance status or access to health care, and not socioeconomic status.

Thank you for your comment. As it was mentioned before, we have focused the paper on the covariates that are reported in the national registry of discharges and avoided any extrapolations to socioeconomic status. We believe that in the revised version of the manuscript we leave all hypothesis that might explain the inequalities found to the Discussion section and make clear that understanding the reasons behind the inequalities requires further investigation.

• This analysis will not be comparable to other studies of social determinants as its groups were defined in ways highly specific to the Chilean population.

We agree with you that our specific results are valid for the Chilean context, in terms of the healthcare organization system. However, we strongly believe the lessons from the Chilean case (implementation of a public policy, such as the GES program, without evaluating its impact) can be applied in any other context. Our work emphasizes the need to tackle the fundamental differences in the population and, the access to screening and high-quality medical care.

Minor points:

- Table 1 would function better as a figure (e.g., line plot) to enable the reader to better evaluate the trends over time.

We were unable to construct a figure with all the info of Table 1. We believe that trends over time are more informative when comparing crude incidence and mortality rates as in Figure 2.

- The results text (e.g., last paragraph in section 3.1) do not make it clear that crude prevalence and mortality are also presented in the figure (though the caption does make this clear).

Following your suggestions, we have updated the figure and make sure the text refers to all the information contained on it.

- Reporting numeric results to two decimal places is standard- Table 2 is particularly hard to follow due to all the decimal places present in it.

Following your suggestions, all numeric results are presented to two decimal places. 

REVIEWER #2: 

Overall

This paper provides important insight into the status of prevalence, incidence, and mortality rates related to colorectal cancer. The authors’ methods seem sound despite the common limitations related to using claims data for this type of research. They use a novel approach in describing the colorectal cancer environment using individual data rather than relying on population level data. My primary concerns regarding the manuscript are described in detail below. Of note, I am not familiar with the health care system in Chile, so please take careful consideration of any comments that may be attributable to my misconceptions of the Chilean health care system, which are solely based on what is presented in the manuscript.

Thank you very much for your insightful comments. We have now made an effort to present the revised version of our manuscript in such a way that readers do not need to have previous knowledge of the Chilean healthcare system or any other specific characteristic of the country.

Primary concerns

1. In the abstract, the authors state that “a national screening program with rapid access to diagnostic and therapeutic procedures is the only way to diminish serious inequality”. This is the strong statement and although these would likely lead to improvements in the observed CRC rates, this statement is not support by evidence in the manuscript.

Thank you for your comment. We have now reorganized the paper in such a way that the focus is determining inequalities in CRC healthcare outcomes such as survival rates. We now leave the potential causes of these inequalities for the Discussion section, where we include the study of differences in screening as one of them. We point out that, as part of our ongoing research, we are studying which are the roots of these differences, and if only in screening (reflected, for example, in later CRC diagnosis in certain groups), then a national screening program might translate into better survival rates. We remark that the focus of the revised version is not to define public policies to decrease these inequalities.

2. The author state that there were 6,626 patients included in the death database and not the treatment database, did the authors consider doing a sensitivity analysis of why those patients were not captured in the treatment database and how their characteristics compared to those who were included in the analyses?

Thank you for your comment. In the revised manuscript, we were able to reduce this number of patients to 6,099. In Section 2, we have included a detailed analysis regarding missing or unavailable data, and how it was incorporated in the study. 

“The constructed death database has 21,734 patients, where 6,099 of them were not in the treatment database with a CRC principal or related hospital discharge between 2001 and 2018, i.e., they died of CRC without receiving treatment requiring hospitalization. Therefore, it is reasonable to assume that the stage of the cancer at the time of diagnosis was advanced enough not to recommend surgery or had comorbidities that prevented them from receiving more invasive treatments. This is also supported by the facts that i) most of these patients (72%) have other causes of hospitalizations, which indicates significant comorbidities that might have contributed to short survival periods after diagnosis, and ii) they are significantly older at death than those with CRC-related discharges (average of 76 versus 70 years old). See Appendix C for details.

Out of these 6,099 patients, 4,398 patients appear in the national registry of inpatient discharges due to hospitalizations for causes different from CRC, and therefore, using this information, we characterized them in terms of health insurance (which is not part of the national death registry). Furthermore, for the 1,701 remaining patients we performed a mean test analysis and determined that their characteristics such as age, sex, and cause of death are statistically the same as the first group of 4,398 patients. Therefore, by assigning a proportionally larger weight to the patients in the first group we performed the incidence and survival analyses correcting for the unknown health insurances.

For the 6,099 patients that died of CRC but did not have any CRC-related discharges, we considered them as diagnosed at the year of death. To further study the robustness of our estimates, we considered two additional scenarios: i) these patients were uniformly diagnosed in the previous 12 months before their deaths, and ii) these patients were diagnosed some time before their deaths, consistently with the survival rates of stage IV CRC cancer patients. For this purpose, we used the survival rates reported in [Arhi et al., Stage at diagnosis and survival of colorectal cancer with or without underlying inflammatory bowel disease: a population-based study].”

3. It is unclear whether covariates were assigned based on first entry into the treatment registry and how changes in insurance status or geographic region were captured over time and how they were addressed in the modeling. Additional description of these methods is recommended.

You have raised an important point, and therefore, we have included a paragraph in the data description to address your concern:

“The type of health insurance, region of residency, and treating hospital were assigned considering the most common value in the dataset. 32,560 patients had the same insurance over the study period, while 35,560 (97% of the dataset) had the same insurance for over 75% of the time. On the other hand, 35,942 patients never changed their region of residence over the study period (98% of the patients), and 35,217 patients were always treated in a hospital in the same region of the patient’s residency. Only 5.4% of the patients received hospital care in a region different from their region of residency, at some point of their treatment.”

4. It is unclear if there is missing data on covariates in the two databases (other than exclusion of unknown sex as described in the methods section) and how it was handled during analyses. The authors should include a description of the distribution of missing data and corresponding methods or provide a statement if there was no missing data. Furthermore, I would recommend further discussion on who may not be captured in the treatment database and how that might affect the reporting of their results.

Thank you for your comment. We fully agree that further explanations were needed. Therefore, in Section 2, we have included a detailed analysis regarding missing or unavailable data, and how it was incorporated in the study.

5. The authors use 2014 as a starting point for the effects of the GES program. I do not know how the program was implemented, but did the authors consider using a lag period to assess the impact of the GES program as its effects might not have had immediate effects during the early implementation period?

You have raised an important point, and therefore, we have included a paragraph in the discussion to address your concern:

“The question remains whether CRC results are not visible because five years is not enough time to assess the impact of the GES program or whether early diagnosis for CRC is the dominant factor that can only be managed with screening programs that are not included in the CRC GES program.” 

6. The authors reference different regions within Chile, however these regions are not noted on any of the included maps. I would recommend that a map be included that note the region numbers for those who are not familiar with the geography of Chile.

Thank for your suggestion. We have now included the corresponding regions in a map, and we presented the regional results from north to south.

7. The authors state differences in incidence and mortality rates based on FONASA public insurance versus privately insured ISAPRES. They use insurance as a proxy for individual socioeconomic position, but do not discuss potential structural or institutional factors related to these insurance practices that may be driving the observed results (i.e., institutional resources, etc.). This is partially discussed in reference to geography, but not insurance status.

Thank you for your comment. As it was mentioned before, we have focused the paper on the covariates that are reported in the national registry of discharges and avoided any extrapolations to socioeconomic status. We believe that in the revised version of the manuscript we have moved all hypotheses that might explain the inequalities found to the Discussion section and make it clear that understanding the reasons behind the inequalities requires further investigation.

8. In reference to geographic differences in CRC rates: Since the findings are based on hospitalization records and not patient residence, how might the observed rates be attributed to population migration or whether individuals travel to certain regions for advanced medical care.

Thank you for your comment. We have now clarified that the analysis was done using patients’ region of residency. Moreover, we have included the number of patients moving to a different region during the study period or getting treatment in a region different from the one of their hometowns. 98% of the patients never changed region of residency and 94.6% of the patients got all their medical care in hospitals located in their region of residency.

9. Cox proportional hazard model results are reported as odds ratios. These should be expressed in hazard ratios.

Following your suggestion, we are presenting hazard ratios for the Cox proportional hazard model.

10. In the discussion, the authors state that the observed geographic gradient parallel ethnic differences in the country. Are the authors implying genetic differences? Socioeconomic factors? Geographical factors? I think the authors should further explain the potential ethic gradient to ensure that undue “blame” is not being placed on potentially minoritized populations when social or structural factors may be primary drivers.

Thank you for your comment. As it was mentioned before, we have focused the revised paper on the covariates that are reported in the national registry of discharges and avoided any extrapolations to socioeconomic status or ethnic distribution. We have now included the complexity of the treating hospitals as one of the covariates, showing that this covariate captures an important part of the survival rates’ disparity. We notice that the geographical distribution of high complexity hospitals is not homogeneous across the country, as it is shown in the map in Appendix D. 

Minor typos

1. Page 10, line 1: “familiar economic consequences…” should be familial.

 Corrected.

2. Figure 2: “Colorrectal” is misspelled. Should be “Colorectal”

 Corrected. 

3. R-squared value for mortality only has 3 significant figures compared to the four significant figures for the other rates.

 Corrected. 

We look forward to hearing from you in due time regarding our submission and to respond to any further questions and comments you may have.

---

## [Decision Letter · Decision Letter 1]

11 Jul 2022

Colorectal cancer trends in Chile: a Latin-American country with marked socioeconomic inequities

PONE-D-22-00874R1

Dear Dr. Mondschein,

We’re pleased to inform you that your manuscript has been judged scientifically suitable for publication and will be formally accepted for publication once it meets all outstanding technical requirements.

Kind regards,

Keith Anthony Dookeran, MD PhD

Academic Editor

PLOS ONE

Reviewers' comments:

Reviewer's Responses to Questions

**Comments to the Author**

1. If the authors have adequately addressed your comments raised in a previous round of review and you feel that this manuscript is now acceptable for publication, you may indicate that here to bypass the “Comments to the Author” section, enter your conflict of interest statement in the “Confidential to Editor” section, and submit your "Accept" recommendation.

Reviewer #1: All comments have been addressed

Reviewer #2: All comments have been addressed

2. Is the manuscript technically sound, and do the data support the conclusions?

Reviewer #1: Yes

Reviewer #2: Yes

3. Has the statistical analysis been performed appropriately and rigorously? 

Reviewer #1: Yes

Reviewer #2: Yes

4. Have the authors made all data underlying the findings in their manuscript fully available?

Reviewer #1: Yes

Reviewer #2: Yes

5. Is the manuscript presented in an intelligible fashion and written in standard English?

Reviewer #1: Yes

Reviewer #2: Yes

6. Review Comments to the Author

Reviewer #1: The authors have done an excellent job responding to my comments, and I am happy to recommend this study for publication.

Reviewer #2: (No Response)

7. PLOS authors have the option to publish the peer review history of their article (what does this mean?). If published, this will include your full peer review and any attached files.

Reviewer #1: No

Reviewer #2: No

---

## [Editor Report · Acceptance letter]

19 Jul 2022

PONE-D-22-00874R1 

Colorectal cancer trends in Chile: a Latin-American country with marked socioeconomic inequities 

Dear Dr. Mondschein:

I'm pleased to inform you that your manuscript has been deemed suitable for publication in PLOS ONE. Congratulations! Your manuscript is now with our production department. 

Kind regards, 

on behalf of

Dr. Keith Anthony Dookeran 

Academic Editor

PLOS ONE